# Immunogenicity Evaluation of Combination Respiratory Syncytial Virus and Varicella–Zoster Virus mRNA Vaccines in C57BL/6J Mice

**DOI:** 10.3390/vaccines13040361

**Published:** 2025-03-28

**Authors:** Ning Luan, Luxia Huang, Jingping Hu, Haihao Zhang, Dandan Gao, Zhentao Lei, Xiaolong Zhang, Han Cao, Cunbao Liu

**Affiliations:** Institute of Medical Biology, Chinese Academy of Medical Sciences and Peking Union Medical College, Kunming 650118, China; luanning@imbcams.com.cn (N.L.); s2023018019@pumc.edu.cn (L.H.); hujingping@student.pumc.edu.cn (J.H.); zhanghh@imbcams.com.cn (H.Z.); ddgao2008@imbcams.com.cn (D.G.); s2023018021@student.pumc.edu.cn (Z.L.); zhangxiaolong@imbcams.com.cn (X.Z.)

**Keywords:** respiratory syncytial virus, varicella–zoster virus, combined vaccines, CMI, mRNA vaccine

## Abstract

Background: Respiratory syncytial virus (RSV) and varicella–zoster virus (VZV) pose significant risks to the elderly and individuals with compromised immune systems. In this study, we investigated whether combining RSV and VZV vaccines could reduce the number of vaccination injections, thereby minimizing discomfort for elderly individuals and reducing manufacturing costs. Methods: In this study, we developed two types of combined RSV and VZV mRNA vaccines. Using RSV and VZV mRNA vaccines administered alone as controls, we evaluated the immune response elicited by the combined mRNA vaccines in C57BL/6J mice. Results: The results demonstrated that RSV mRNA, VZV mRNA, and a mixture of both could be effectively encapsulated in lipid nanoparticles (LNPs) with uniform particle sizes. Compared to the administration of either the RSV or VZV mRNA vaccine alone, the delivery of two kinds of mRNA LNP combination formulation—whether directly mixed or encapsulated two mRNAs in the same LNP formulation—elicited comparable IgG titers, neutralization titers, cell-mediated immunity (CMI), and CD4^+^ T-cell responses. Conclusions: In conclusion, this study establishes the feasibility of combining RSV and VZV mRNA-LNP vaccines, laying a solid foundation for clinical trials of combined RSV and VZV vaccines.

## 1. Introduction

Respiratory syncytial virus (RSV) is the predominant pathogen that causes severe lower respiratory tract disease (LRTD) in infants, young children, and elderly individuals [1]. Data indicate that there are 336,000 hospitalizations and 14,000 in-hospital fatalities annually, representing a significant burden of LRTD among individuals over 65 years of age [2,3]. Furthermore, the incidence and impact of RSV in older adults are likely underestimated because of infrequent testing [4]. Varicella–zoster virus (VZV), which remains dormant in the ganglia once infection occurs, can lead to herpes zoster (HZ) and trigger postherpetic neuralgia (PHN) in elderly immunocompromised individuals [5]. In the United States, one in three people may experience shingles in their lifetime, with the incidence increasing after age 50. For older adults, the treatment options for RSV-related LRTD and VZV-related PHN are insufficient, emphasizing the need for the development of VZV and RSV vaccines.

There are several similarities between RSV and VZV vaccine studies. Initially, elderly people and immunocompromised individuals are populations that are sensitive to virus attacks, and vaccination could benefit them most. Conversely, the importance of cell-mediated immunity (CMI) has been emphasized in the development of both vaccines [6]. The success of mRNA vaccines has been demonstrated by COVID-19 mRNA vaccines, including BNT162b2 and mRNA-1273 [7,8]. Compared with traditional vaccines, the immune characteristics of the RSV vaccine and VZV vaccine, which require the induction of CMI, align well with the principles of mRNA action that induce strong T-cell responses, indicating that an mRNA platform could be a good candidate for VZV and RSV vaccine development.

The leading VZV vaccine product on the market is Shingrix^TM^, a subunit vaccine developed by GlaxoSmithKline (GSK). Studies have demonstrated that Shingrix^TM^; offers over 90% protection efficacy in individuals aged 50 to 59 years and those over 70 years and also significantly reduces the risk of PHN [9,10]. However, the AS01B adjuvant used in Shingrix^TM^; has a production limit that affects the supply of the vaccine. Once it was developed, the mRNA vaccine platform emerged as a promising platform for next-generation VZV vaccine development, with cost-effectiveness and strong CMI [11]. Compared with Shingrix^TM^, mRNA-1468, which was developed by Moderna and has a short gE sequence, shows promising immunogenicity, regardless of whether 100–200 μg of mRNA is injected once or 50 μg of mRNA is injected twice [12]. Another study has shown that ZOSAL, a VZV mRNA vaccine, also induces superior vaccine immunity over Shingrix^TM^; in mice and rhesus macaques [13]. Currently, there are three vaccines on the market for RSV: Arexvy from GSK, Abrysvo from Pfizer, and mRNA-1345 from Moderna [14,15]. The first two are subunit vaccines, and the last one is an mRNA vaccine. There are still several RSV mRNA vaccines in preclinical trials, some of which are combined vaccines. For example, SP0256 from Sanofi is an RSV-hMPV-PIV triple vaccine that is currently undergoing phase 1 clinical evaluation. IN006, a bivalent RSV mRNA vaccine from Innorna, encodes two proteins with stabilized prefusion conformation F proteins, RSV-A and RSV-B, and is the first RSV mRNA vaccine to receive clinical trial approval in China. The long-term protection rates of combined mRNA vaccines for RSV and others are still under investigation.

Fusion (F) glycoproteins play crucial roles in viral fusion, budding, and dissemination of RSV [16]. Stabilization of the prefusion (pre-F) conformation has recently been shown to be a superior antigen that can elicit strong immune responses [17]. Glycoprotein E (gE), one of the most abundant VZV glycoproteins, plays essential roles in virus replication and transmission between ganglia cells [18,19]. The mutations Y569A, S593A, S595A, T596A, and T598A in the C-terminus of gE (gE-M) have been demonstrated to localize gE to the trans-Golgi network or plasma membrane [20]. Our previous study revealed that, compared with other forms of gE, gE-M can induce superior immunogenicity in VZV-mRNA vaccines [19,21]. In this study, an mRNA-based Pre-F RSV vaccine and an mRNA-based gE-M VZV vaccine were constructed. By directly mixing two LNP-mRNA vaccines or combining the two vaccines via LNP encapsulation, we obtain two forms of combined mRNA vaccines. C57BL/6J mice were immunized to evaluate their immunological responses. Antibody titers, the release of Th1-oriented cytokines, and the percentage of CD4^+^ T-cells that express Th1-oriented cytokines were tested. The results revealed that there were no significant differences between the combined vaccine groups and the vaccine-alone groups. To date, no study has investigated combined immunization strategies for VZV and RSV. Administering combination vaccines has advantages such as simplifying the immunization process, reducing the number of vaccine doses, increasing adherence among vaccine recipients, lowering vaccination costs, improving vaccine coverage, and increasing willingness to vaccinate. This study demonstrates the practicability of combining RSV and VZV vaccines, which may alleviate the discomfort of vaccination in elderly individuals to some extent and has promising development prospects.

## 2. Materials and Methods

### 2.1. Preparation of mRNA Vaccines

Sequences of Pre-F mRNA and gE-M mRNA were codon-optimized and synthesized by Sangon Biotech (Shanghai, China). The details of the VZV mRNA sequence can be found in our previous report and was named gE-M-P [21]. Additionally, the sequence of Pre-F mRNA was the same as that of mRNA-1345 (Moderna Inc., Cambridge, MA, USA) [22]. Briefly, nucleic acid sequences of 5′-UTR, 3′-UTR, and poly(A) tail were obtained from the Pfizer/BioNTech BNT162b2 mRNA vaccine [23], which has been demonstrated to have strong immunogenicity. For in vitro mRNA synthesis, T7 polymerase-mediated DNA-dependent RNA transcription with a T7 High Yield RNA Transcription Kit was used (N1-Me-Pseudo UTP) (Vazyme Biotech Co., Ltd., Nanjing, China). Furthermore, post-transcriptional mRNAs were purified using VAHTS RNA Clean Beads (Vazyme Biotech Co., Ltd., Nanjing, China), resulting in total quantity of 200 μg mRNA per sample. LNP-mRNA vaccines were prepared via a modified encapsulation procedure as previously described [24]. Briefly, lipids (from AVT Medical Technologies Ltd, Shanghai, China) were dissolved in ethanol at a molar ratio of 46.3: 9.4: 42.7: 1.6 (MC3: DSPC: cholesterol: DMG-PEG2000). The lipid mixture was combined with 0.1 M of citric acid–citrate solution (pH 4.0) containing mRNA at a ratio of 3:1 (water:lipid) using a microfluidic mixer (Precision Nanosystems, Vancouver, BC, Canada). Then, the mRNA-LNP products were dialyzed in a 40× volume of PBS using 50 kDa of MWCO centrifugal filtration tubes (Millipore, Burlington, MA, USA) and centrifuged at 3000 rpm. The hydrodynamic diameters and polydispersity index (PDI) were measured using a Malvern ZEN3600 device (Malvern Instruments Ltd., Worcestershire, UK). mRNA encapsulation was verified using 1% denatured agarose gels, and the mRNA nucleic acid load was measured with a Quant-it^TM^; RiboGreen RNA Assay Kit (Invitrogen, Hillsboro, OR, USA). Parts of samples were lysed by adding the same volume of 1% Triton and incubated overnight at 4 °C, and the encapsulation rate was calculated using the following fomula: (the mass of lysed samples/the mass of initial nucleic acid load) × 100%. All LNP-formulation vaccines were stored at −20 °C and melted in the ice before use.

### 2.2. Animal Studies

Female C57BL/6J mice (6 weeks old, 15–18 g) were purchased and maintained by the Central Animal Services of the Institute of Medical Biology, Chinese Academy of Medical Sciences (IMB, CAMS). One week before immunization, the mice were randomly divided into groups (*n* = 6) and maintained under SPF conditions. Briefly, there were five groups: the RSV LNP-mRNA vaccine group [Group Rv]; the VZV LNP-mRNA vaccine group [Group Vv], the RSV LNP-mRNA vaccine and VZV LNP-mRNA vaccine group [Group Rv+Vv], the RSV mRNA and VZV mRNA were coencapsulated in LNPs group [Group (R+V)v], and the PBS control group.

Mice were immunized twice in 4-week intervals. The immunogens (50 µL) were injected intramuscularly (i.m.) into the thigh muscles. Two weeks after the final immunization, mice were sacrificed via overdose injection of 300 mg/mL tribromoethanol. Blood samples were collected via cardiac puncture, and spleens were dissected for further analysis.

### 2.3. Antibody Titer Detection

Whole blood was clotted at 4 °C overnight, and immunized sera were collected following centrifugation at 3000 rpm for 20 min. Two kinds of immunogens, which were supplied by AtaGenix Laboratory (Wuhan, China), were used to analyze the antibody titers of the different groups. The RSV fusion protein (Pre-F) was used for Group Rv, Group Rv+Vv, Group (R+V)v, and PBS. The VZV glycoprotein gE was used for Group Vv, Group Rv+Vv, Group (R+V)v, and PBS. Briefly, the immunogens were coated on 96-well plates (NEST Biotechnology Co., Ltd. Shenzhen, China) at a concentration of 2 µg/mL. After incubation overnight at 4 °C, the plates were blocked with 5% (*w*/*v*) skim milk at 37 °C for 1 h. Two serial dilutions of mouse serum were added and incubated at 37 °C for 1 h. Bound antibodies were detected using a goat anti-mouse IgG-horseradish peroxidase (HRP) conjugate (1:10,000, Biolegend, Hercules, CA, USA) as a secondary antibody. Ten min after the addition of the substrate 3,3′,5,5′-tetramethylbenzidine (TMB, Solarbio, Beijing, China), 2 M of H_2_SO_4_ was added to terminate the reaction. The absorbance at 450 nm was determined with a spectrophotometer (BioTek Instruments, Inc., Winooski, VT, USA). Immunoglobulin G (IgG) titers were defined as end-point dilutions with cutoff signals above OD450 = 0.15, and IgG titers lower than 200 were set at 200 for calculations.

### 2.4. RSV Cytopathic Effect Neutralization Test (CPENT)

The capacity of immune sera to neutralize RSV was assessed through neutralizing antibody titration using the RSV A2 strain (ATCC, VR-1540) and HEp-2 cells (ATCC, CCL-23). The 50% inhibitory dilution (ID50) was defined as the serum dilution at which the cytopathic effect (CPE) was reduced by 50% in comparison with that in the virus control wells. In brief, the RSV A2 strain was incubated with a series of dilutions of the test samples (a 50-fold dilution as a starting point, with four subsequent 4-fold dilutions) in 96-well plates for 1 h at 37 °C. Subsequently, 10,000 HEp-2 cells were added to each well. Following a 4–7-day incubation period in a 5% CO_2_ environment at 37 °C, crystal violet solution (100 μL per well) was added, and the mixture was incubated at room temperature for 30 min. The wells were then rinsed with ddH_2_O, and CPE was observed and recorded. The neutralizing antibody titers were calculated by summing the number of positive and negative CPE wells using the Reed–Muench method.

### 2.5. Isolation of Splenocytes

Mouse spleens were dispersed with a 40 µm cell strainer (Beijing Labgic Technology Co., Ltd. Beijing, China). The cells were harvested via centrifugation at 800× *g* for 5 min. Then, ammonium chloride potassium lysis buffer was added for another 5 min at room temperature. After red cell lysis, the cells were re-harvested via centrifugation at 800× *g* for 5 min. The splenocytes were counted and resuspended in Roswell Park Memorial Institute (RPMI) 1640 medium supplemented with 10% *v*/*v* fetal bovine serum (FBS) (both from VivaCell Biosciences, Shanghai, China) and 1% penicillin–streptomycin (Thermo Fisher, Hillsboro, OR, USA) at a final concentration of 1 × 10^7^ cells/mL. The cell culture dishes/plates and centrifuge tubes were obtained from NEST Biotechnology Co., Ltd. (Wuxi, China).

### 2.6. Enzyme-Linked Immunosorbent Assay (ELISA)

For the ELISA, each well of a 96-well plate (NEST Biotechnology Co., Ltd., Shenzhen, China) was seeded with 100 µL of splenocytes at a final concentration of 1 × 10^6^ cells/well. Similarly, immunostimulants (gE or Pre-F, 10 µg/mL) were then added to each well, along with an equivalent volume of PMA+ ionomycin (Dakewe Bioengineering Co., Ltd., Shenzhen, China) as a positive control. After incubation for 24 h at 37 °C, the supernatants were collected for cytokine level determination. Briefly, unconjugated anti-IL-2 (3 µg/mL) and anti-IFN-γ (4 µg/mL) antibodies, dissolved in PBS, were coated onto 96-well plates at 4 °C overnight. Subsequently, the plates were blocked with 1% (*w*/*v*) BSA in PBS at 37 °C for 1 h. Samples were added to each well and incubated for 3 h at room temperature. Biotin-conjugated antibodies against IL-2 and IFN-γ (2 µg/mL, Invitrogen, CA, USA), dissolved in 1% BSA, were then added and incubated for another 1h, followed by the addition of HRP-conjugated streptavidin (1 µg/mL, Biolegend, Hercules, CA, USA), which was incubated for 30 min. TMB substrate and 2 mol/L sulfuric acid were sequentially added, and the absorbance at 450 nm was measured using a SYNERGY 4 microplate reader (BioTek Instruments, Inc., Winooski, VT, USA).

### 2.7. Enzyme-Linked Immuno-Spot (ELISpot) Assay

Clear 96-well plates (Millipore, Burlington, MA, USA) were precoated with unconjugated anti-IL-2 (2 µg/mL) and anti-IFN-γ (2 µg/mL) antibodies (Invitrogen, CA, USA). Splenocytes were resuspended in serum-free medium for universal ELISpot (DAKEWE, Shenzhen, China) and added to each well at a final concentration of 3 × 10^5^ cells/well. Immunostimulants (gE or Pre-F protein) at a final concentration of 20 µg/mL were added to the splenocytes, which were subsequently incubated overnight. The next day, an ELISpot assay kit (BD, USA) was used according to the manufacturer’s protocol. Spots were counted with an ELISpot reader system (Autoimmun Diagnostika GmbH, Strassberg, Germany) after immunoimaging.

### 2.8. Flow Cytometry

Splenocytes were prepared and cultured in 24-well plates at a density of 2 × 10^6^ cells/well. To induce immune responses, immunostimulants (gE or Pre-F) at a final concentration of 10 µg/mL were added and incubated for 2 h. Brefeldin-A (5 µg/mL) was added to block cytokine release, followed by overnight incubation. Cells were harvested and stained with Zombie NIR^TM^; dye for 20 min at room temperature to differentiate live cells from dead ones. To prevent nonspecific binding to Fc receptors, cells were treated with CD16/CD32 antibodies (5 µg/mL) and incubated at 4 °C for 10 min. Subsequently, PerCP/Cyanine 5.5-tagged anti-mouse CD4 antibody was applied, and the mixture was kept at 4 °C for an additional 30 min. After fixation with 4% formaldehyde at room temperature for 20 min, the cells were washed twice with permeabilization wash buffer and stained with PE-tagged anti-mouse IFN-γ and APC-tagged anti-mouse IL-2 antibodies for 1 h at room temperature. Finally, more than 20,000 CD4^+^ cell events were analyzed using a Cyto-FLEX flow cytometer (Beckman, Indianapolis, IN, USA) and FlowJo software (V10.6.2, BD, Franklin Lakes, NJ, USA).

### 2.9. Statistical Analysis

GraphPad Prism 9.5.0 (GraphPad Software Inc., La Jolla, CA, USA) was used for the statistical analyses. The data were analyzed using one-way analysis of variance (ANOVA) followed by Tukey’s multiple comparisons test, in which the mean of each column is compared with the mean of every other column. Notably, the PBS group was not included in any comparisons. Additionally, it should be noted that there is one missing data point in the Vv group because of an identified outlier.

## 3. Results

### 3.1. LNPs Efficiently Encapsulated mRNA Antigens with Homogeneous Particle Sizes

The mRNAs of RSV and VZV can be efficiently transcribed in vitro (IVT) from the corresponding DNA sequences. The IVT products were verified using agarose gel electrophoresis, revealing a band for RSV of approximately 1500 base pairs (bp) (Figure 1A, line 1) and a band for VZV of approximately 2000 bp (Figure 1A, line 2). Three forms of LNPs were prepared: LNPs encapsulating RSV mRNA alone (Rv), VZV mRNA alone (Vv), and LNPs co-encapsulating both RSV and VZV mRNAs [(R+V)v]. All the encapsulated LNP-mRNA vaccines exhibited good integrity according to the denatured agarose gel; the LNPs protected the mRNAs, preventing their migration in the gel (Figure 1A, lines 4–6), whereas the bands from the lysed samples corresponded to the locations of the RSV and VZV IVT mRNA products (Figure 1A, lines 7–9). The hydrodynamic diameters of these nanoparticles were 85.10 ± 0.82, 85.56 ± 0.69, and 84.91 ± 1.14 nm, respectively (Figure 1B). The polydispersity indices (PDIs), which indicate the homogenous population of the samples on the basis of size, were approximately 0.15 for all formulations (Rv: 0.159, Vv: 0.164, and (R+V)v: 0.144, Figure 1C), suggesting good uniformity of the nanoparticles.

Moreover, the encapsulation efficiency exceeded 80% (85.2% for Rv, 82.1% for Vv, and 83.6% for (R+V)v, Figure 1D). With an initial quantity of 8 μg/dose of mRNA products, the final injection quantities for the groups were 6.82 μg/dose for Rv, 6.57 μg/dose for Vv, 13.39 μg/dose for Rv+Vv, and 13.38 μg/dose for (R+V)v. Overall, the results presented in Figure 1 demonstrate that LNPs effectively encapsulated the mRNA vaccines, resulting in uniform particle sizes, stable structures, and complete encapsulation of LNP-mRNA nanoparticles.

### 3.2. Combination Vaccine Groups Achieved Antibody Titer Levels Similar to Those of Single Vaccine Injection Groups

Two methods were used to assess RSV antibody titers. The mean Pre-F IgG antibody titers for groups Rv, Vv+Rv, and (V+R)v were 256,000, 138,667, and 170,667, respectively (Figure 2A). The titers of the combined vaccine groups (Vv+Rv and (V+R)v) were slightly lower than those of the RSV mRNA-LNP group (group Rv), but no significant differences were observed. The neutralizing antibody titers of the RSV vaccine groups were determined using the RSV-A2 strain (Figure 2B). The results indicated that group (V+R)v had the highest neutralizing antibody titer at 3597. Similarly, there were no significant differences between the groups. It has been reported that CMI, rather than IgG titers, affects the immunology of VZV vaccines. Here, the VZV-gE IgG titers were also routinely assessed (Figure 2C). The mean gE IgG antibody titers for groups Vv, Vv+Rv, and (V+R)v were 256,000, 277,333, and 256,000, respectively. In summary, the use of mRNA-LNP vaccines in different combinations did not affect humoral immunity compared with the use of mRNA-LNP vaccines alone.

### 3.3. Groups in Which the Combination Vaccine Was Injected Presented Slightly Increased CMI

The levels of two cytokines in the splenocyte supernatants, IL-2 (Figure 3A,C) and IFN-γ (Figure 3B,D), which are indicative of Th1 activity, were measured using ELISA. In the RSV-Pre-F groups (Figure 3A,B), the average concentrations of IL-2 and IFN-γ were 600.6 ng/mL and 167.0 ng/mL, respectively, in the Vv+Rv group and 537.8 ng/mL and 302.5 ng/mL, respectively, in the (V+R)v group. These values were slightly higher than those observed in the Rv group (345.0 ng/mL and 137.6 ng/mL), although no significant differences were noted. In the VZV-gE M groups (Figure 3C,D), the average values of IL-2 and IFN-γ were 547.8 and 119.2 ng/mL, respectively, in the Vv+Rv group and 654.3 and 730.0 ng/mL in the (V+R)v group, both of which were slightly greater than those in the Vv group (379.0 and 120.2 ng/mL). However, again, no significant differences were found.

Similar trends were observed in the ELISpot results (Figure 4). In terms of the average number of IL-2 spots (Figure 4A,B), the groups receiving the combination vaccine presented a greater number of spots than the groups receiving the vaccine alone. In the RSV groups, the number of spots was 123.2, 102.8, and 37.3 in the Vv+Rv, (V+R)v, and Rv groups, respectively. In the VZV groups, the number of spots was 82.2, 110.7, and 65.2 in the Vv+Rv, (V+R)v, and Vv groups, respectively. The IFN-γ spots (Figure 4C,D) were significantly weaker than those for IL-2 (Figure 4A,B). In the RSV groups, the number of spots was 113.8, 158.5, and 13.3 in the Vv+Rv, (V+R)v, and Rv groups, respectively. In the VZV groups, the number of spots was 74.5, 60.7, and 14.6 in the Vv+Rv, (V+R)v, and Vv groups, respectively.

On the basis of the ELISA and ELISpot analyses, the combined use of vaccines can be concluded to induce elevated levels of CMI against RSV and achieve comparable levels of CMI to VZV.

### 3.4. Compared with the LNP-mRNA Vaccine Group, the Combined LNP-mRNA Vaccine Group Presented Slightly Fewer CD4^+^ T-Cells

LNP-mRNA vaccines have self-adjuvant characteristics and can stimulate robust humoral and cell-mediated immunity [25]. CD4^+^ T-cells are important indicators of CMI. Flow cytometry was used to assess the percentage of IL-2/IFN-γ-producing CD4^+^ T-cells among the splenocytes of the mice in the different experimental groups. In all the examination results (Figure 5A–D), the vaccine groups receiving LNP-mRNA in combination presented slightly lower CD4^+^ T-cell counts than those receiving the LNP-mRNA vaccine alone; however, these differences were not statistically significant. The trends observed via flow cytometry were opposite those observed via the ELISA and ELISpot assays, which may be attributed to the fact that the number of IL-2/IFN-γ-producing CD4^+^ T-cells does not align with that of the IL-2/IFN-γ-producing splenocytes. Furthermore, this phenomenon may also have resulted from the stimulation time for flow cytometry being 2 h, whereas it was 24 h and 16 h for ELISA and ELISpot, respectively.

## 4. Discussion

Combination vaccines include multivalent and multicomponent vaccines. The development of combination vaccines has the advantage of reducing the number of required injections and, hence, effectively enhancing immunization coverage [26,27,28]. For example, the development of a diphtheria, tetanus, pertussis (DTaP)-based multivalent vaccine has significantly increased DTaP immunization coverage [26]. Another study of the DTaP combination vaccine in the United States showed that children receiving combination vaccines had higher rates of completing the full four-dose immunization schedule and on-time immunization than those receiving a single vaccine. Due to the characteristics of maximized vaccination coverage and improved patient convenience, combination vaccines have wide economic prospects. Based on the characteristics of RSV and VZV, combination vaccines are promising and necessary to develop and could best benefit the elderly population. Here, the immunogenicities of two forms of combined RSV and VZV mRNA vaccines were first evaluated. Our study demonstrated that RSV and VZV mRNA vaccines could be combined to achieve the following effect: “one shot can protect against both RSV and VZV”.

If the antibody response levels generated after vaccination with the combined vaccine are not comparable to those produced by administering individual vaccines, the efficacy of the vaccine should be considered. In terms of the antibody titers, the combined vaccines in this study did not have this problem, and the IgG antibody titers of the combined vaccine groups and the single vaccine groups were comparable (Figure 2A,C). The neutralizing antibody titers of the RSV-related vaccine groups against strain RSV-A were tested and did not significantly differ between the groups (Figure 2B).

Previous studies have demonstrated that CMI responses, rather than antibody titers, are pivotal in the prevention and control of initial VZV infection, along with the reactivation of latent infection. Studies investigating RSV have highlighted that a robust, RSV-specific CD4^+^ T-cell response is linked to protective immunity against LRTD, particularly in relation to the immune mechanisms that underlie vaccine efficacy outcomes [29]. Various T-cell subsets have been characterized on the basis of their cytokine profiles. Type 1 cytokines (IL-2 and IFN-γ) promote CMI and enhance the cytotoxicity of macrophages against intracellular pathogens [30]. A type 1 cytokine profile has been associated with better clinical outcomes in terms of vaccine immunogenicity. In this study, the secretion levels of IL-2 and IFN-γ, the numbers of IL-2- and IFN-γ-producing cells, and the percentages of IL-2- and IFN-γ-expressing CD4^+^ splenocytes were measured using ELISA (Figure 3), ELISpot (Figure 4), and flow cytometry (Figure 5), respectively. The results revealed that there was no difference in the Th1-oriented CMI response between the combined vaccine groups and the vaccine-alone groups. All groups elicited a robust RSV pre-F or VZV gE-specific immune response that was at least tenfold greater than that at baseline (the PBS group), suggesting that immunization could be ample for generating protective immunity [31].

One difficulty in RSV vaccine development is the protection rate against multiple RSV pandemics. On 26 June 2024, the US Centers for Disease Control and Prevention (CDC) reported that the efficacy of mRNA-1345 in preventing RSV-LRTD in older adults decreased to 50.3% within two RSV epidemic seasons; however, no long-term protection data were revealed. The data from one single administration of the RSV subunit vaccines produced by Pfizer (Abrysvo) and GSK (Arexvy) are 48.9% (the efficacy against two or more RSV-LRTDs in a median follow-up period of 13.9 months [32]) and 67.2% (the efficacy against two or more RSV-LRTDs across two complete RSV seasons [33]), respectively. The development of RSV mRNA vaccines with high and long-term protection rates still has a long way to go. The nucleic acid sequence of an open reading frame (ORF) is related to proper antigen folding, the ability of mRNAs to reduce the secondary structures of GCs, and the translational rates of antigens. However, there is a growing body of evidence concerning the regulatory roles played by UTRs in terms of the stability of nucleic acid molecules and translation efficiency [7,34]. Our previous studies demonstrated that screening for matching UTR sequences is a necessary strategy for the development of mRNA vaccines [19,21]. In this study, ORF sequences were derived from Pre-F, which is used in the mRNA-1345 vaccine, and gE-M, which is used in the mRNA-1468 vaccine for RSV and VZV, respectively. The noncoding sequences, including UTRs, the 5′ cap, and the poly(A) tail, were sourced from Pfizer BNT162b2 [34]. Both the mRNA-LNP vaccines used in this study, whether used alone or in combination, exhibited good uniformity, integrity, and immunogenicity. We propose that further studies are needed to validate whether modifications in UTRs, the different forms of combined mRNA vaccines, and changes in the immune schedule of combined mRNA vaccines prolong the duration of immune protection against RSV.

## 5. Conclusions

Our findings demonstrate that mixing the two encapsulated LNP-mRNA vaccines for RSV and VZV (Group Rv+Vv) can achieve desirable results. Similarly, encapsulating RSV mRNA and VZV mRNA together in LNPs (Group (R+V)v) can lead to similarly admirable immune responses. Both combined approaches can reduce production and testing costs, confer protection and adequate response, and reach the goal of “one shot to protect against both RSV and VZV”. Furthermore, combining both vaccines does not result in serious adverse immune responses in C57BL/6J, which justifies a clinical trial for further combined immunization.

## Figures and Tables

**Figure 1 vaccines-13-00361-f001:**
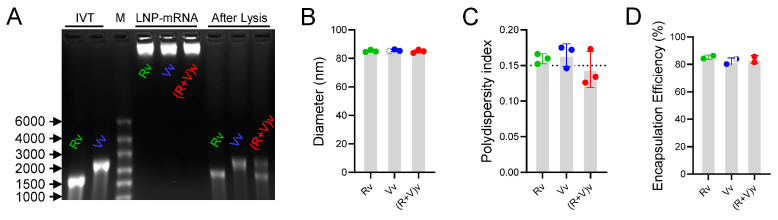
Characterization of LNP-mRNA vaccines. (**A**) The loaded mRNA was visualized using 1% denatured agarose gel electrophoresis. Band 1 and band 2: the IVT products of RSV mRNA and VZV mRNA. Band 3: DL5000 marker. Bands 4–6: encapsulated mRNA-LNPs. Bands 7–9: the lysis products of the mRNA-LNPs. (**B**) The diameters of the LNPs were tested by a size analyzer. (**C**) Polydispersity index (PDI) of LNPs; the dashed line indicates PDI = 0.15. (**D**) Encapsulation efficiency of loaded mRNA. IVT, in vitro transcription. Rv, RSV; Vv, VZV; (R+V)v, RSV+VZV. Samples were replicated three times; data are shown as mean ± SD.

**Figure 2 vaccines-13-00361-f002:**
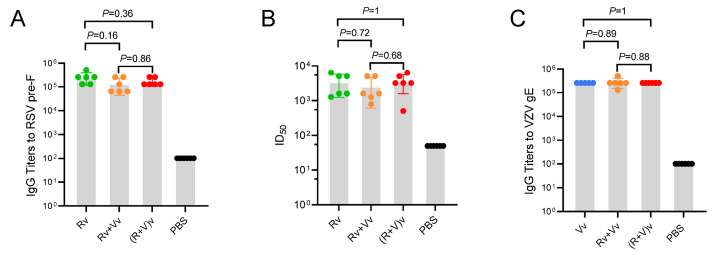
Antibody titers of immunized mouse sera. (**A**) Pre-F-specific IgG titers detected using enzyme-linked immunosorbent assay (ELISA). (**B**) The 50% inhibitory dilution (ID50) of RSV vaccine-related compounds was assayed in Hep-2 cells. (**C**) gE-specific IgG titers detected by ELISA. The data were compared using one-way analysis of variance (ANOVA) followed by Tukey’s multiple comparisons test. Each point represents an individual mouse.

**Figure 3 vaccines-13-00361-f003:**
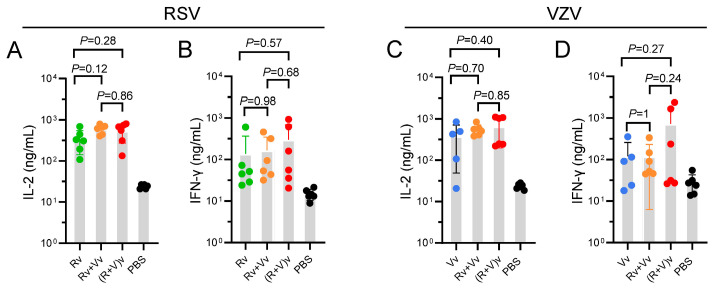
The levels of secreted IL-2 and IFN-γ in the splenocyte population were assayed using ELISA. (**A**,**B**) RSV vaccine-treated groups stimulated with Pre-F. (**C**,**D**) VZV vaccine-treated groups stimulated with gE. One-way analysis of variance (ANOVA) followed by Tukey’s multiple comparisons test was used, and there were no significant differences between groups.

**Figure 4 vaccines-13-00361-f004:**
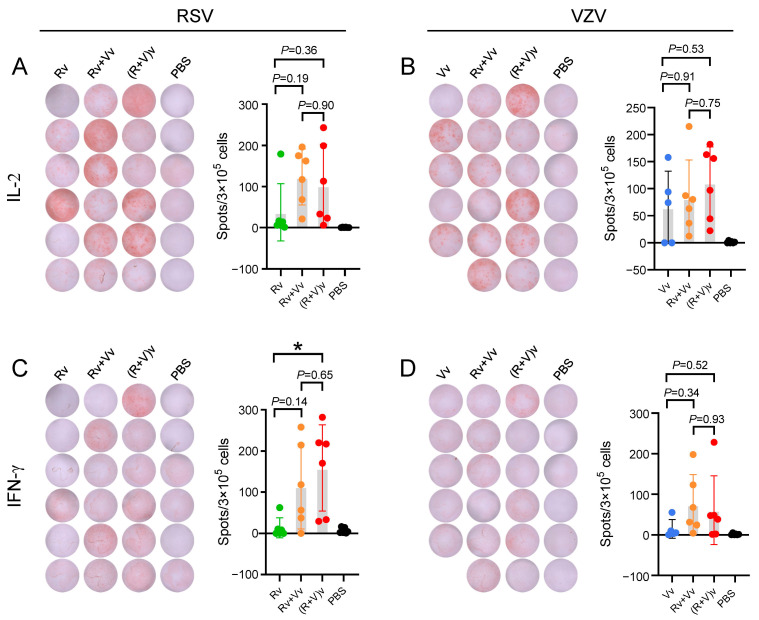
The numbers of IL-2- and IFN-γ-secreting splenocytes were measured using ELISPot. (**A**,**B**) show the levels of IL-2 produced by splenocytes; (**C**,**D**) show the levels of IFN-γ. Pictures of individual spots are also shown. Means were compared using one-way ANOVA followed by Tukey’s multiple comparisons test. Points represent individual mice, * *p* < 0.05.

**Figure 5 vaccines-13-00361-f005:**
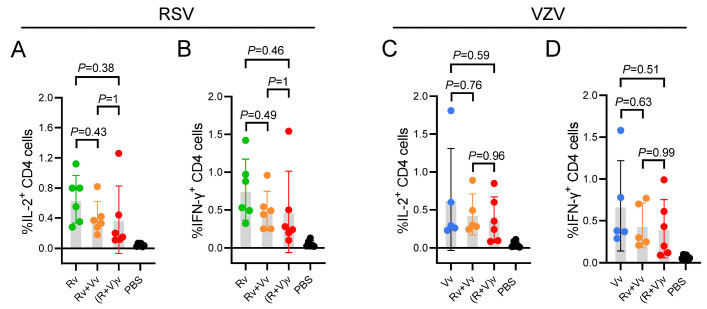
Flow cytometry assay for IL-2/IFN-γ-producing CD4^+^ T-cells. (**A**,**B**) Proportion of IL-2- and IFN-γ-producing CD4^+^ T-cells among the RSV vaccine-treated groups; (**C**,**D**) Proportion of IL-2- and IFN-γ-producing CD4^+^ T-cells among the VZV vaccine-treated groups. The data were analyzed using one-way ANOVA followed by Tukey’s multiple comparisons test.

## Data Availability

All data used during the study are available from the corresponding author upon request.

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
