# Peer review of "Immunogenicity Evaluation of Combination Respiratory Syncytial Virus and Varicella–Zoster Virus mRNA Vaccines in C57BL/6J Mice"

_vaccines, 2025, doi:10.3390/vaccines13040361_

Round 1

Reviewer 1 Report

Comments and Suggestions for Authors

In this manuscript, Luan et al, addressed the “Immunogenicity evaluation of combination RSV and VZV mRNA vaccines in C57BL/6J mice”. Having examined the manuscript, I note that though it discusses interesting observations, to be considered for MDPI Vaccines, the following are some of the comments that the authors might find useful for future submission. This manuscript is well-structured and delivers insightful information regarding the potential of the combined use of RSV and VZV mRNA-LNP vaccines is important for reducing manufacturing costs, minimizing discomfort from vaccinations for elderly individuals, and providing a foundation for future studies on combined RSV and VZV vaccines. This type of studies are extremely valuable for the scientific community at a global level.

 Reviewer Comments

  1. The authors estimated total IgG immune responses in serum. It is insufficient to fully characterize the vaccine induced humoral immune responses. If feasible, I suggest authors to include serum antibody responses such as IgG subclasses (IgG1, IgG2a, IgG2b, IgG3) provide critical information regarding Th1 and Th2 anti RSV and anti VZV  immune responses
  2. The authors evaluated CD4+ T cell responses by measuring IL-2 and IFN-γ, which offer insights into T helper cell activation and Th1 immunity. However, this assessment is incomplete and does not provide a full picture of cellular immunity. Given that mRNA-lipid nanoparticle vaccines are known to elicit robust CD8+ T cell responses. If feasible, I suggest authors to assess CD8+ T cell activation and their cytokine secretion, which plays a critical role in viral clearance and long-term protection.
  3. What is the stability of vaccine formulations. The authors did not provide any data regarding the stability mRNA-LNP vaccine formulations, which is important because LNP vaccine formulations are sensitive to degradation during its storage.
  4. The authors did not evaluate the dendritic cell activation and antigen presentation in RSV and VZV mRNA LNP vaccines. Inclusion of this data would further strengthen the manuscript.
  5. The authors has administered the vaccines via intramuscular (IM) route, which is primarily induces systemic immunity but is less effective in generating mucosal immunity. However, since, RSV and VZV are the viruses mainly infect mucosal surfaces, evaluation of mucosal immune responses (IgA and IgG) would provide insights of mucosal immunity. If feasible, include this data to the manuscript.
  6. Authors need to correct typographical error throughout the manuscript 

Author Response

Thank you for your professional comments. Please check our responses to your comments as follows:

  1. Thanks for your advice. Generally, mRNA vaccines will induce Th1-oriend immune response. We feel sorry that we didn’t set Alum adjuvant groups as Th2-oriend immune response control, thus the detection of other IgG subclasses (IgG1, IgG2a, IgG2b, IgG3) would be meaningless that incomparable between mRNA vaccine groups. We will consider more thoughtfully in the future work.
  2. As our experience, CD8+ T cell responses couldn’t be tested by flow cytometry and get a desirable result (PMID: 35631559): data normally located lower than 0.04% and cannot be distinguished clearly, which result in no significant differences can be detected between groups. On the one hand, the reason may be the method we used is in the low resolution., On the other hand, the sacrificed time of mice, which also the spleen collected time, may not the correct time for the enrichment and detection CD8+ T cells, as long CD8+ T cells are long-lasting immune cells.
  3. Once made, the vaccine formulations were stored at -20℃ before use. We illustrate it in line 118-119 in the revised manuscript. LNP formulation made by our lab is stable in this storage method at least two months, no matter the physical properties and immunization effects, data were not shown.
  4. Thanks for your thoughtful recommendation, your suggestion will made our results more reliable and get more internationally recognition. The in vitro effects of mRNA vaccines, such as dendritic cell activation, the antigen presentation, and the quantity of antigen expressed will be considered in the future work to strengthen the clinical application of combined RSV and VZV vaccines.
  5. As we known, IgA and IgG of mucosal immune responses can only be detected in the BALF, throat swab, nasal swab, or vaginal swabs, et.al. I’m sorry that we didn’t collect those mucosal samples when mice sacrificed, thus IgA and IgG of intramuscular route couldn’t be administrated in this study. We will consider more sincerely in the future experimental design.
  6. We have checked the revised manuscript carefully to make sure no longer typographical error will exist.

Reviewer 2 Report

Comments and Suggestions for Authors

1. Line 231. Please report the standard deviation. Also, the proper term is hydrodynamic diameter. 
2. Line 234. “homogenous population” would be a better term.
3. Line 235. It is not clear how the encapsulation efficiency was measured. Please describe it better.
4. Figure 1. Please add the number of replicas performed to obtain these graphs.
5. Figures 2, 3, 4,  and 5. I find it unconventional that first, an ANOVA was done and then a K-W. Normally, a test to determine normality is done (ANOVA is not a normality test), and then, depending on the result, either an ANOVA or a K-W is performed. Also, if a K-W was performed, what post hoc test was used? K-W only tells you if there are differences within a population; it is the post hoc test that gives you the actual p numbers between groups. Also, the convention of p > 0.05 is to say that it is not significant (n.s.) instead of reporting the p-value. finally, as a note p goes from 1 to 0 so p > 0.99 is 1.
6. Figure 2. Please show the IgG titers to RSV pre-F for the Vv group and the gG titers to VZV gE for the Rv group. The same observation for Figure 2b.
7. Figure 3 is confusing because the scales are different; please use log scales so the combined groups can be compared between Fig 3A and 3C and Figu 3B and 3D. The same goes for figures 4 and 5 (in 5, only fix the scale of 5a, so it goes from 0 to 2).
8. Line 364 is extremely confusing. What do you mean by the RSV pandemic affecting RSV vaccine development?
9. Throughout the manuscript, there is the claim that both mRNAs are co-encapsulated. However, there is no evidence of that. When you have both mRNAs, the liposomes are formed; some can contain Rv, others Vv, and finally, ones Rv + Vv. Hence, I think a better claim is that mixing both mRNAs with the lipid preparation has the same outcome as mixing two independent liposome populations. I know that this is a matter of wording, but this article is great, and thus, it will only improve it by being careful. 
10. Line 390. There is no data to support the claim that “encapsulating 389 VZV mRNA and RSV mRNA together in LNPs (Group (R+V)v) can lead to even better  immune responses.” The (R+V)v and Rv + Vv formulaitons gave the same results. Nonetheless, the important result is being missed! Combining both vaccines confers protection and adequate response, and, more importantly, this study shows that combining both vaccines does not result in serious adverse immune responses, thus justifying a clinical trial.

Author Response

  1. In Line 237 of revised manuscript, term “diameter” has been superseded by “hydrodynamic diameter”. And standard deviation of each data has been added.
  2. It has been changed in line 239 of revised manuscript.
  3. The method of measuring encapsulation efficiency has been re-described in line 116-118 for better understand.
  4. the number of replicas has been added in Figure 1 legend. (line 260-261).
  5. As mentioned in parts” Statistical analysis: The data were analyzed using one-way analysis of variance (ANOVA) followed by Tukey’s multiple comparisons test, in which the mean of each column is compared with the mean of every other column”. What’s described in figure 2/3/4/5 legends: “The data were compared using one-way analysis of variance (ANOVA) followed by the Tukey’s multiple comparisons test” were mistakes, and has been revised in the manuscript.

Generally, it is described as not significant (n.s.) when P>0.05. While “ns” is a common result between groups in this study, we choose to put the p value on each comparison to make reader more clearly with the variations between groups. What’s more, we changed the statement of “p > 0.99” with “p=1”, including Figure 2C, Figure 3D, Figure 5A, and Figure 5B.

  1. The results of IgG titer to RSV pre-F for the Vv group, the IgG titer to VZV gE for Rv group, and the neutralization titer to RSV A2 strain for the Vv group are negative.
  2. Thank you for your professional advice, I changed the scale to Log scales in Figure 3, which make it more pleasing to read. While I didn’t use log scale in figure 4, because of some value would down below the X axis. In figure 5, I re-image Figure 5A make the Y axis goes from 0-2.
  3. The statement has been revised in Line 386. What we want to express is “One difficulty in RSV vaccine development is the protection rate to multiple RSV pandemic”.
  4. Thank you for your suggestion. The idea you mentioned is a brilliant view that we haven't considered before. Although two mRNAs were encapsulated in the LNP, but whether the formed LNPs contains Vv, Rv, or (R+V)v, and in what percentage was unknown. It needed to be further examined, to make our study more standing.
  5. Thanks for your kindly suggestion. The conclusion part was re-paraphrase as follows in line 411-418: Our findings demonstrate that mixing the two encapsulated LNP-mRNA vaccines for RSV and VZV (Group Rv+Vv) can achieve desirable results. Similarly, encapsulating RSV mRNA and VZV mRNA together in LNPs (Group (R+V)v) can lead to similarly admirable immune responses. Both combined approaches can reduce production and testing costs, confers protection and adequate response, and reach the goal “one shot to protect against both RSV and VZV”. What’s more, combining both vaccines does not result in serious adverse immune responses in Balb/C, which justifying a clinical trial for further combined immunization.